Peer**J**

# Prediction uncertainty assessment of a systems biology model requires a sample of the full probability distribution of its parameters

Simon van Mourik[1,2], Cajo ter Braak[1], Hans Stigter[1] and Jaap Molenaar[1,2]

[1] Biometris, Wageningen University and Research Center, Wageningen, The Netherlands
[2] Netherlands Consortium for Systems Biology, Amsterdam, The Netherlands

## ABSTRACT

Multi-parameter models in systems biology are typically 'sloppy': some parameters or combinations of parameters may be hard to estimate from data, whereas others are not. One might expect that parameter uncertainty automatically leads to uncertain predictions, but this is not the case. We illustrate this by showing that the prediction uncertainty of each of six sloppy models varies enormously among different predictions. Statistical approximations of parameter uncertainty may lead to dramatic errors in prediction uncertainty estimation. We argue that prediction uncertainty assessment must therefore be performed on a per-prediction basis using a full computational uncertainty analysis. In practice this is feasible by providing a model with a sample or ensemble representing the distribution of its parameters. Within a Bayesian framework, such a sample may be generated by a Markov Chain Monte Carlo (MCMC) algorithm that infers the parameter distribution based on experimental data. Matlab code for generating the sample (with the Differential Evolution Markov Chain sampler) and the subsequent uncertainty analysis using such a sample, is supplied as Supplemental Information.

Corresponding author
Simon van Mourik,
simon.vanmourik@wur.nl

## INTRODUCTION

By combining experiments and mathematical model analysis, systems biology tries to unravel the key mechanisms behind biological phenomena. This has led to a steadily growing number of experiment-driven modeling techniques (*Klipp et al., 2008*; *Stumpf, Balding & Girolami, 2011*). Useful models provide insight and falsifiable predictions and the dynamic models used in systems biology are no exception. A complicating factor in multi-parameter models is that often many parameters are largely unknown. Even among similar processes parameter values may vary multiple orders of magnitude, and experimentally they are often hard to infer accurately (*Maerkl & Quake, 2007*; *Buchler & Louis, 2008*; *Teusink et al., 2000*).

A standard procedure for parameter estimation is via a collective fit, i.e., estimating the values of the unknown parameters simultaneously by fitting the model to time series data, resulting in a calibrated model. This approach has to cope with several obstacles, such as measurement errors, biological variation, and limited amounts of data. Another often met problem is that different parameters may have correlated effects on the measured dynamics leading to highly uncertain or even unidentifiable parameter estimates (*Zak et al., 2003a*; *Zak et al., 2003b*; *Raue et al., 2009*). Systems biology models are typically 'sloppy' in that some parameters or combinations of parameters are well defined, whereas many others are not (*Brown & Sethna, 2003*; *Gutenkunst et al., 2007b*). It was argued on the basis of this sloppiness that collective fits are more promising for obtaining useful predictions than direct parameter measurements, and that "modelers should focus on predictions rather than on parameters" (*Gutenkunst et al., 2007b*). One might expect that sloppiness and the associated parameter uncertainty automatically lead to uncertain predictions, but that is not necessarily so. For example, in the literature a model has been reported with 48 parameters calibrated to only 68 data points and with highly uncertain parameter values, but the predictions were quite accurate (*Brown et al., 2004*).

In this paper we fit an illustrative model and a diverse set of six systems biology models from the BioModels database (*Li et al., 2010*) to different amounts of (simulated) time series data. For each model, we study the prediction uncertainty of a number of predictions by a full computational uncertainty analysis. We measure the uncertainty in the predicted time course by the dimensionless quantifier $Q_{0.95}$ which has the nice property that $Q_{0.95} < 1$ indicates tight predictions, whereas $Q_{0.95} \geq 1$ implies uncertainty in the dynamics that is likely to obscure biological interpretation. Using this quantifier, we show that some models allow more accurate predictions than others, but, more importantly, that the uncertainty of the predictions may greatly vary within one model. We argue that prediction uncertainty assessment must therefore be performed on a per-prediction basis using a full computational uncertainty analysis (*Savage, 2012*). We indicate how such an analysis can be performed in a relatively easy way. It requires a sample representing (strictly speaking, approximating) the distribution of the parameters of the model. The importance of such a sample (an ensemble of parameter sets) for uncertainty analysis has been pointed out earlier (*Brown & Sethna, 2003*; *Gutenkunst et al., 2007a*), but we go one step further and recommend modellers to make available not only the mathematical model and its typical parameter values and operating conditions, but also a sample representing the distribution of its parameters (Fig. 1). For other reasons, such as verification and reproducibility of the research, data and software to obtain such a sample should also be made available.

Within the Bayesian framework (*Jaynes, 2003*; *MacKay, 2003*) this is a sample as generated by an MCMC algorithm (*Robert, Marin & Rousseau, 2011*) that collectively fits the parameters of the model to experimental data. We advocate the Bayesian framework as it allows inclusion of prior information (based on prior parameter knowledge, general biological considerations (*Grandison & Morris, 2008*) and direct parameter measurements) to be combined with the information contained in time series data. If no prior information

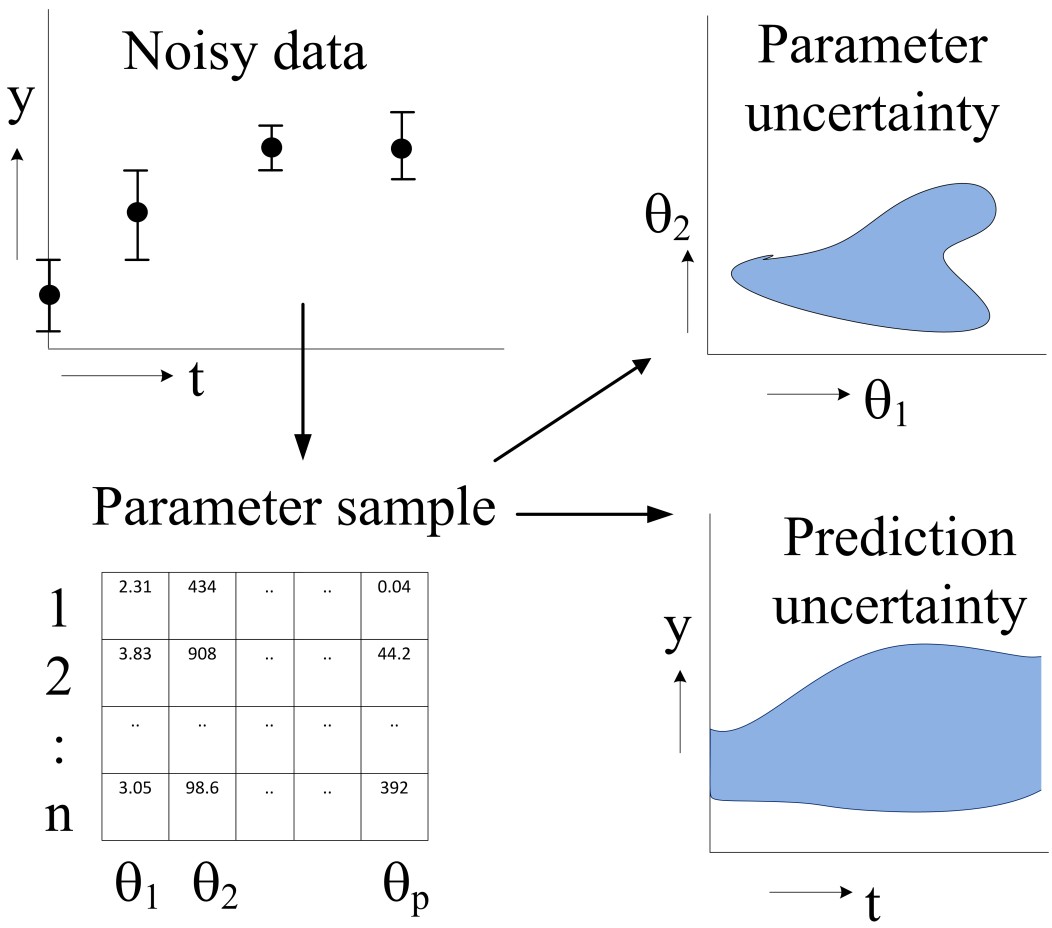

**Figure 1** **Flowchart of parameter estimation and uncertainty analysis.** The model dynamics are fitted to noisy time series data, typically by an MCMC algorithm, resulting in a sample of parameter values representing their posterior distribution. Typical applications of this distribution are the computation of a credible region of parameters (showing parameter uncertainty) and full computational uncertainty analysis of predictions.

is available, a noninformative prior can be constructed. For a discussion on prior selection, see *Robert, Marin & Rousseau (2011)*.

It is useful at this point to compare the Bayesian approach with the frequentist approach of prediction profile likelihood (PPL) (*Kreutz et al., 2013*) which is another approach to uncertainty analysis. PPL requires heavy computation for uncertainty analysis of each prediction. In particular, it requires the solution of a large number of nonlinear optimization problems, each under a different nonlinear constraint. By contrast, the Bayesian approach requires (perhaps very) heavy computation to obtain a sample representing the distribution of the parameters. But once this sample has been obtained, the uncertainty analysis is easy, namely by calculating the prediction for each parameter vector in the sample and summarizing the so obtained set of predictions. For this reason, the Bayesian framework has strong appeal when uncertainty analysis is required for

**Peer**J

many predictions from a single model, as in the current paper. For other aspects in the comparison between the Bayesian and frequentist approaches see *Raue et al. (2013)*.

Parameter uncertainty and prediction uncertainty play an important role also in hydrology (*Beck, 1987*; *Beven & Binley, 1992*), ecology (*Omlin & Reichert, 1999*), and meteorology (*Hawkins & Sutton, 2009*). What we contribute here is a quantifier for the uncertainty in predicted systems dynamics that expresses the uncertainty in a concise way, and the focus on what is needed to carry out a full computational uncertainty analysis, namely samples representing all sources of uncertainty. In the internet age, such samples can easily be made available for prospective users of the model.

## MATERIALS AND METHODS

In this section we present the methodology to sample the probability distribution of the parameters, and to define the uncertainty of predicted dynamics. To illustrate how this works in practice, we have implemented the used sampling algorithm together with a prediction uncertainty computation algorithm in Matlab$^®$ software, together with a sample of the parameters (see the Supplemental Information).

### Model class

We consider models that are formulated in terms of differential equations and have the following form:

$$\dot{x}(t) = f(x(t), \theta, u(t))$$
$$y(t) = g(x(t), \theta)$$
$$x(0) = h(\theta). \tag{1}$$

The dynamics of the internal state vector $x$ is a function $f$ of parameter vector $\theta \in \mathbb{R}^p$, with initial condition $x(0)$, and external input vector $u(t)$. The output $y \in \mathbb{R}^m$ is a positive function $g$ of the internal state and $\theta$. To obtain the output time series $y(t)$, we have to numerically integrate $\dot{x}(t) = f(x(t), \theta, u(t))$, which is the time consuming part of the solution.

### Estimation of the posterior distribution of the parameters

We adopt the Bayesian framework, in which a prior distribution of the parameters is combined with data to form the posterior distribution $\pi(\theta)$ of the parameters, according to Bayes' theorem

$$\pi(\theta) = p(\theta|y_d) = \frac{p(\theta)p(y_d|\theta)}{p(y_d)}. \tag{2}$$

Here $p(\theta)$ denotes the density of the prior distribution of the parameters based on prior knowledge and $p(y_d|\theta)$ the likelihood of the measured data $y_d$ given $\theta$. The denominator, $p(y_d)$, is the marginal likelihood of the data $y_d$ and acts as a $\theta$-independent normalization constant which is not relevant in MCMC algorithms. We assume that the output $y(t_i)$, $i = 1, \ldots, n$ contains uncorrelated Gaussian noise, with variance $\sigma_i^2$ per time point $i$, so the

likelihood of the measured data $y_d$, given $\theta$, is

$$p(y_d|\theta) = \prod_{i=1}^{n} \frac{1}{\sqrt{2\pi\sigma_i^2}} \exp\left(-\frac{(y_i(\theta) - y_{d,i})^2}{2\sigma_i^2}\right), \tag{3}$$

with $y_i(\theta)$ the model output at time $t_i$ given $\theta$. Inserting (3) into (2) and taking the logarithm gives the log-posterior

$$\log(\pi(\theta)) = c - \frac{1}{2}\chi^2(\theta) + \log(p(\theta)), \tag{4}$$

with $c$ a constant independent of $\theta$, and $\chi^2$ a measure of the fitting error:

$$\chi^2(\theta) = \sum_{i=1}^{n} \frac{(y_i(\theta) - y_{d,i})^2}{\sigma_i^2}. \tag{5}$$

The penalized maximum likelihood parameter $\theta^{PML}$ maximizes (4). Draws from the posterior $\pi(\theta)$ are obtained by an MCMC algorithm, which, starting from a user-defined initial value of $\theta$, generates a stochastic walk through the parameter space. In iteration $k$ of the walk, a new candidate solution $\theta$ is proposed based on the current solution $\theta_k$. We use symmetric proposal distributions centered at $\theta_k$. In this case the proposed $\theta$ is accepted or rejected using the Metropolis acceptance probability $\min(1, r)$, where $r = \frac{\pi(\theta)}{\pi(\theta_k)}$. So, the more probable a parameter vector is with respect to the data, the more probable it is to be accepted. In this paper the proposals for the walk are generated by the DE-MCz algorithm, which is an adaptive MCMC algorithm that uses multiple chains in parallel and exploits information from the past to generate proposals (*ter Braak & Vrugt, 2008*). As consecutive draws are dependent, it may be practical to reduce the dependence by thinning, that is, by storing every $K$th draw (with $K > 1$). After a number of iterations (the burn-in period) the chain of points $\{\theta_k\}$ will be stationary distributed with a local density that represents the posterior $\pi(\theta)$. The burn-in iterations are discarded.

## A quantifier of uncertainty in predicted systems dynamics

Since we want to compare the uncertainty in predictions of time courses over different time intervals and for different models, we need a quantifier that is independent from model specific issues, such as the number, and the dimensions of the variables. Also, biologists are generally interested in percentage difference rather than absolute difference, so the quantifier should be independent of the typical order of magnitude. To that end we introduce a measure for prediction uncertainty that satisfies these requirements, but we do not claim that this quantifier is unique. However, our present choice has the advantage that it allows a nice interpretation as will be explained below. We first define a measure for the prediction uncertainty of one component of the output and then average the outcomes over the components.

For a predicted output $y_p(t, \theta)$ on time interval $[0, T]$, the quantifier $Q$ represents the relative deviation of $y_p(t, \theta)$ from the penalized maximum likelihood prediction $y_p(t, \theta^{PML})$, integrated over time:

$$Q_i(\theta) = \frac{1}{T} \int_0^T \left( \log_b \left( \frac{y_{p,i}(t, \theta)}{y_{p,i}(t, \theta^{PML})} \right) \right)^2 dt, \qquad (6)$$

with $y_{p,i}$ the $i$th component of $y_p$, with $i = 1, \ldots, m$, and $Q = \frac{1}{m} \sum_i Q_i$. $Q$ integrates differences over time on a logarithmic scale with base $b$, and is therefore symmetric with respect to relative over- and underestimations (as opposed to, e.g., a sum of squared errors). The underlying assumption is that $y$ is always positive, which holds true for all models describing concentration dynamics. When only a few time points are of biological interest, the integrand in (6) may be approximated by a summation. The base $b$ characterizes the order of magnitude of the discrepancies that $Q$ is sensitive to. For example, if $\frac{1}{b} < \frac{y_{p,i}}{y_{p,i}^{ML}} < b$ holds for all time points, this will result in $Q < 1$, while a few points outside this range will quickly result in $Q > 1$. In this paper we use $b = 2$, so only differences of a factor two or higher can result in $Q > 1$. The choice of $b$ represents the maximum magnitude of deviations that a prediction is allowed to have, so it should in general be selected based on biological grounds.

If $\{\theta\}$ is the collection of sampled $\theta$'s reliably representing density $\pi(\theta)$, then the density $\pi(Q(\theta))$ is reliably represented by $Q(\theta)$ with $\theta \in \{\theta\}$ (*Robert, Marin & Rousseau, 2011*). We denote the level $\alpha$ prediction uncertainty with $Q_\alpha$, the $\alpha$ quantile of the distribution of $Q$. $Q_\alpha$ is therefore the deviation of a prediction relative to the penalized maximum likelihood prediction, at confidence level $\alpha$ ($\alpha$-level deviation). We use $\alpha = 0.95$ throughout.

## Algorithm to estimate prediction uncertainty

The algorithm for estimating prediction uncertainty naturally falls apart in two sub-algorithms. Part I deals with estimating parameters by exploiting the prior knowledge, the model, and the data. This first part yields the sample of parameter values representing their posterior distribution. Part II is the focus of this paper, and performs the full computational uncertainty analysis by taking as input the sample of parameter values from the first part and by calculating the prediction for each member of this sample. Part I is computationally more intensive than Part II. Because prospective users of the model need to carry out only Part II, it is essential that the sample of parameter values obtained in Part I is stored and made available. In full:

Part I: Estimation of the posterior parameter distribution
  1. To calculate the posterior $\pi(\theta)$ in Eq. (2), we use Eq. (3) for the likelihood, together with a log-uniform prior for the parameters, that is, a uniform prior for the logarithm of the parameters.
  2. A collection of solutions $\{\theta\}$ is generated by MCMC sampling; in this paper we use DE-MCz (*ter Braak & Vrugt, 2008*). This collection constitutes the sample approximating $\pi(\theta)$.

Part II: Computational uncertainty analysis

1. $Q(\theta)$ is computed for each $\theta \in \{\theta\}$. The $\alpha$ level prediction uncertainty $Q_\alpha$ is approximated by taking the largest $Q$ after discarding the $100\alpha\%$ largest $Q(\theta)$ values.

2. For visualization of the uncertainty in the predicted systems dynamics, for each $\theta \in \{\theta\}$ the output $y(t)$ is calculated. For each time point the $\frac{1}{2}(1-\alpha)100\%$ largest and smallest predicted values of $y(t)$ are discarded, whereafter the minimum and maximum values are plotted, creating an envelope of predicted dynamics.

A structural difference between $Q$ and the visualization is that $Q$ integrates the deviation over the total function $y(t)$, while the visualization displays prediction uncertainty per time point and per component of $y(t)$. An alternative to definition (6) is to define $Q$ as the deviation of $\log_b(y_p(t,\theta))$ from the average log-prediction, thus replacing $\log_b(y_p(\theta^{PML}))$ with $\overline{\log_b(y_{p,i}(\theta))}$.

## RESULTS

### Illustrative example

To demonstrate what kind of effects can be expected when studying prediction uncertainty of models with parameter uncertainty, we use a model example that represents a highly simplified case of self-regulation of gene transcription:

$$\dot{y}(t) = \frac{10y(t)^2}{\theta_1 + y(t)^2} - \theta_2 y(t) + y(t)u(t). \tag{7}$$

Here, $y(t)$ represents protein concentration and $\dot{y}(t)$ its rate of change. Changes in $y(t)$ stem from self-regulation (the first term in the equation, which is a Hill function *Alon, 2006*), decay (the second term), and an input function (the third term) given by $u(t) = \sin(t)$. The input represents some external signal, e.g., a triggering mechanism (*van Mourik et al., 2010*). The unknown parameters involved in self-regulation and decay are $\theta_1$ and $\theta_2$, respectively. They are to be estimated from time series data, for which we used simulated data at ten time points (Fig. 2A). We assumed independent Gaussian noise on each measurement, with a mean of 10% of the $y$ value, and a log-uniform prior for the parameter distribution. We estimated $\theta_1$ and $\theta_2$ by generating 1000 draws from their distribution using MCMC sampling. This yielded a sample of 1000 draws. From this sample we derived a 95% credible region of $\theta_1$ and $\theta_2$ (Fig. 2B) and also point-wise 95% credible intervals (envelopes) for three sets of predictions (Fig. 2C). The prediction sets differ in starting value $y(0)$ or the input function $u(t)$. The envelopes for prediction (Fig. 2C) are obtained by inserting each of the 1000 draws into the (perturbed) system, calculating the prediction on a fine grid of time points by evaluating the (modified) differential equation (7) and summarizing the 2.5 and 97.5 percentiles of the predictions at each time point. Details are given in the Methods section.

The data give much more information about $\theta_2$ than about $\theta_1$ (Fig. 2B, note the different scales on the axes), that is, the dynamics of the time series is sensitive to changes in $\theta_2$, but not sensitive to changes in $\theta_1$; the model thus shows sloppiness. The reason is not hard to

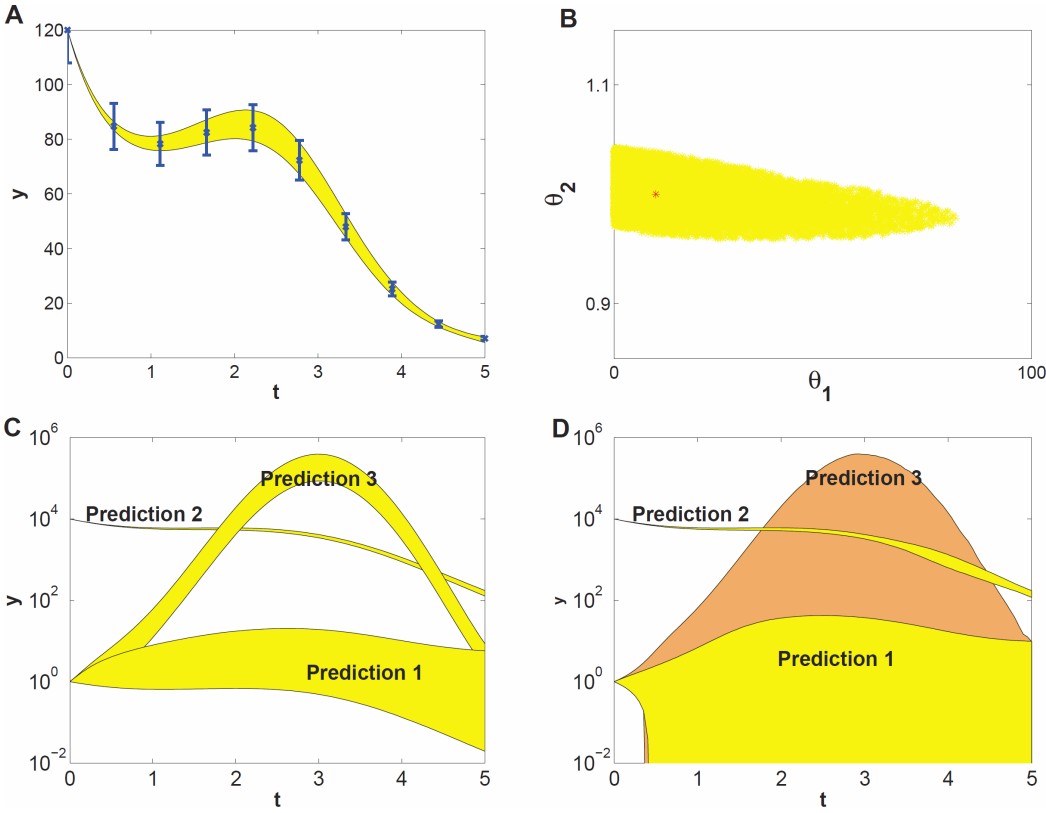

**Figure 2 Prediction uncertainty following from parameter uncertainty and computational methods.** (A) The model dynamics fitted to the time series data. The 95% uncertainty region is displayed in yellow, the error bars denote the standard deviation of the measurement noise. (B) Parameter uncertainty represented by the 95% credible region, containing the true parameter vector (red ⋆). (C) Predicted dynamics for different initial concentrations (Predictions 1 and 2), and for a 7 fold amplitude increase of the sinusoidal input (Prediction 3). (D) Predicted dynamics via local sensitivities (linearized covariance analysis). The uncertainty of Prediction 3 is plotted in a different shade and includes the uncertainty region of Prediction 1.

understand: since we have used a high initial value, most of the $y(t)$ curve has values much bigger than $\theta_1 = 10$, so that the parameter $\theta_1$ has little influence on the dynamics (see Eq. (7)). In more complex models, credible regions may have far more complicated shapes than in this illustrative example.

The prediction sets Prediction 1 and Prediction 2 (Fig. 2C) are obtained by starting the system for each draw of the sample at a low initial concentration ($y(0) = 1$) and at a very high concentration ($y(0) = 10^4$), respectively. The uncertainty in prediction increases in time for Prediction 1 and is much higher than that for Prediction 2. The predicted dynamics is thus highly sensitive to parameter uncertainty for Prediction 1, but fairly insensitive for Prediction 2. Based on Predictions 1 and 2 one might guess that the uncertainty is related to the magnitude of $y(t)$, but this guess is wrong as the third prediction set shows. In this set, we predict the time course starting at the same low value as in Prediction 1 but now the input amplitude is increased 7-fold ($u(t) = 7\sin(t)$). This

drives the concentration dynamics into an oscillating motion which can be predicted rather precisely (Fig. 2C), since now the input $u(t)$ dominates the dynamics.

We also estimated prediction uncertainty via a local sensitivity analysis, namely linearized covariance analysis (LCA, see the Supplemental Information for details). LCA extrapolates parameter- and prediction uncertainty from a second- and first order Taylor expansion, respectively, and does not take into account the higher order terms that in this case constrain the credible region and the predictions, but note that linearization can also lead to predictions that appear more precise than they really are. Consequently, LCA dramatically overestimates the uncertainties of Predictions 1 and 3 (Fig. 2D).

This simple example illustrates that prediction uncertainty depends on the type of prediction and is hard to foresee intuitively. We cannot evaluate the prediction uncertainty on the basis of some simple guidelines after only inspecting the credible region of the parameters. Because of the nonlinearities in the system, the credible region is untruthful for evaluating predictions and their uncertainty (*Savage, 2012*). These results suggest that a full computational uncertainty analysis, as performed here (Fig. 2C), has always to be done in order to estimate the consequences of parameter uncertainty for prediction uncertainty. The key element in this analysis is the sample representing the probability distribution of the parameters.

It is convenient to quantify prediction uncertainty, that is, the variability among predicted time courses. We do so on the basis of the quantity $Q$ defined in Eq. (7) in the Methods section, which expresses the deviation between two time courses, one being calculated for an arbitrary set of parameter values and the other using the penalized maximum likelihood estimator of the parameters. In our simulation studies, the latter is replaced by the true parameter values. The quantity $Q$ is calculated for each draw of the parameters. The higher the $Q$ value of a prediction, the higher its prediction uncertainty is. Each prediction set yields 1000 values of $Q$, which are summarized by their 95% percentile, indicated by $Q_{0.95}$. $Q_{0.95}$ values smaller than 1 indicate tight predictions. For example, the $Q_{0.95}$ values for Predictions 1, 2 and 3 in Fig. 2C are 18, 0.02 and 1.5, respectively. Even for this simple model the uncertainty among different predictions thus varies by a factor 900 which is an order of 2.9 on logarithmic scale.

## Scanning a variety of systems biology models

Next, we study prediction uncertainty for six models from the BioModels database (*Li et al., 2010*). We selected these models to represent some cross section of systems biology models; The systems they represent greatly differ in numbers of variables, parameters, and the types of equations. The parameters in these models represent process rates. The models include circadian rhythm, metabolism, and signalling (*Laub & Loomis, 1998*; *Levchenko, Bruck & Sternberg, 2000*; *Tyson, 1991*; *Goldbeter, 1991*; *Vilar et al., 2002*; *Leloup & Goldbeter, 1999*), see the Supplemental Information for details. For each model, a data set is created in the same manner as described for the simple model above, with parameter values and initial conditions as provided in the database. Each data set was analyzed using an MCMC sampling algorithm yielding a sample of 1000 draws representing the distribution of the

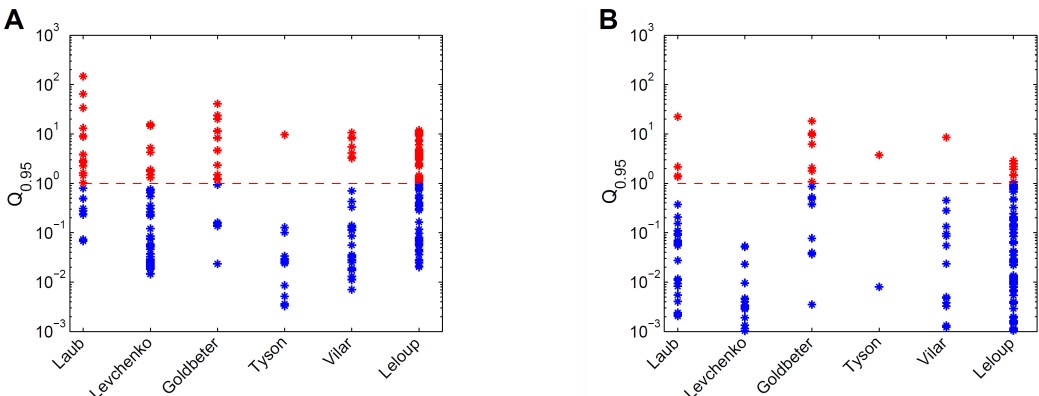

**Figure 3 Comparison of prediction uncertainties for six different models.** (A) Uncertainties of predictions of six different models, expressed in terms of $Q_{0.95}$. A blue * denotes a tight prediction ($Q < 1$), and a red * denotes an uncertain prediction ($Q \geq 1$). (B) Prediction uncertainties obtained after reducing parameter uncertainty by a ten-fold decrease in data noise.

parameters. This number turned out to be large enough for our prediction purposes as checked in the Supplemental Information. Subsequently, different sets of predictions were made. For each prediction set, the system was perturbed and the time course of the perturbed system was predicted on the basis of each of the 1000 draws of parameter values, whereafter prediction uncertainty was quantified on the basis of $Q_{0.95}$. Perturbations consisted of artificially increasing and decreasing each parameter in turn by a factor of 100 (different factors yielded similar results, see Supplemental Information). For a model with $p$ parameters, we so obtain $2p$ different perturbations, prediction sets, and $Q_{0.95}$ values.

For the six models the prediction uncertainty is displayed in terms of the $Q_{0.95}$ quantifiers (Fig. 3). For the collective fits we used a data set with 10 data points per variable, with an assumed Gaussian noise of $\sigma_i = 0.1 y_{d,i}$ per data point $i$. (Fig. 3A). The sizes and locations of the $Q_{0.95}$ distribution ranges considerably differ per model. For example, the Tyson model gives mostly precise predictions, and the Laub model gives mostly uncertain predictions. Next, the accuracy of the data was increased artificially by decreasing the noise with a factor 10, and observed the effect on the $Q_{0.95}$ distribution ranges (Fig. 3B). With more accurate data, the $Q_{0.95}$ values are reduced, but not in a very systematic way. For the Vilar model, for example, all predictions have become precise, except for one. More importantly, for all six models the $Q_{0.95}$ values scatter enormously, in ranges that span 2 to 4 orders of magnitude. There is no clear connection between the number of parameters and the maximum $Q_{0.95}$ values or the size of the ranges. By consequence, prediction uncertainty is not a characteristic of a calibrated model, but must be evaluated on a case-by-case basis.

## DISCUSSION

By introducing the quantifier $Q_{0.95}$ for prediction uncertainty we were able to show that prediction uncertainty is typically not a feature of a calibrated model, but must be calculated on a per-prediction basis using a full computational uncertainty analysis.

The adjective *full* stresses that the analysis must use (a representative sample of) the distribution of the parameters and not only the 95% confidence region or credible region of the parameters. The reason is that the parameter values inside the confidence region may be outliers in terms of the prediction, just because of the nonlinearities of the model (*Savage, 2012*). The key element is thus a faithful representation of the remaining uncertainty in the parameters after fitting the model, in practical terms represented by a sample of parameter values. This sample looks like a dataset with parameters as columns and draws from the distribution as rows (Fig. 1). We used a sample of 1000 nearly independent draws obtained by thinning the MCMC chain. The sample is an approximation of the full posterior distribution of the parameter values. The reliability of this approximation is a point that deserves attention. We checked this by varying the size of the sample and observing whether the results for the quantifier Q showed good convergence. For the models studied in this paper, we found that this convergence was indeed reached if we used a sample size of 1000 points, but for other cases it could happen that this number must be larger.

The value of $Q_{0.95}$ is interpretable. If $Q_{0.95} \geq 1$, the uncertainty is so high that it may obscure biological interpretation of a model prediction, whereas this is unlikely if $Q_{0.95} < 1$. When the uncertainty indeed causes predictions to be biologically ambiguous, $Q_{0.95}$ could be incorporated as a design criterion in experimental design, i.e., additional experiments can be designed or selected to minimize $Q_{0.95}$ (*Vanlier et al., 2012a*; *Vanlier et al., 2012b*; *Apgar et al., 2010*).

We adopted the Bayesian framework in our analysis. This framework has the advantage that in the collective fits prior information on the parameters can be incorporated, such as from general biological knowledge (*Grandison & Morris, 2008*) and previous experiments. The prior in a Bayesian analysis is also a means to ensure that a parameter attains no physically unrealistic values, when fitting the model to data. Current Bayesian MCMC algorithms naturally lead to a sample of the posterior distribution that represents the remaining uncertainty in the parameters. Alternatively, bootstrap sampling (*Efron & Tibshirani, 1993*) could perhaps be used to create such a sample. MCMC can also be applied to stochastic models. At each iteration the stochastic model has to be run again, so that averaging/summarizing over the chain averages/integrates over the stochastic elements in the model. As this simple approach often leads to very low MCMC acceptance rates, it is advantageous to treat the stochastic elements as additional unknowns in the MCMC with a Metropolis-within-Gibbs approach (*ter Braak & Vrugt, 2008*). Particle MCMC is a promising method in this area in which proposals for the additional unknowns are made via a particle filter (*Andrieu, Doucet & Holenstein, 2010*; *Vrugt et al., 2013*). Intersubject variability is a simple form of a stochastic model in which only the parameters of the model vary among subjects. The populations of models approach (*Britton et al., 2013*), which is related to the Generalized Likelihood Uncertainty Estimation (GLUE) approach (*Beven & Binley, 1992*) that is popular in hydrology, is a way to deal with such variability. The formal Bayes approach (and thus MCMC) can also be used. Moreover, it can disentangle

inter-subject variability (which cannot be reduced) and uncertainty (which can in principle be reduced by collecting more data) as demonstrated for a pharmacokinetic model in *ter Braak & Vrugt (2008)*. A comparison of formal Bayes and GLUE can be found in *Vrugt et al. (2009)*.

Of course there are computational issues in the Bayesian approach which increase with the number of the parameters and the complexity of the model. *Gutenkunst et al. (2007a)* speeded up their Monte Carlo analysis (a random walk Metropolis algorithm) by periodically updating the step size based on the current Fisher Information Matrix, but such an updating scheme is generally not recommended as it may change the stationary distribution of the chain (*Roberts & Rosenthal, 2007*). We used the adaptive MCMC algorithm DE-MCz (*ter Braak & Vrugt, 2008*), that automatically adapts to the size and shape of the posterior distribution. DE-MCz allows parallel computation when the chains have access to a common store of past-sampled parameter values, and is designed to quickly find and explore very elongated subspaces in the posterior distribution of the parameters, which are due to sloppiness. Such subspaces point to problems with parameter identifiability. An alternative approach to explore these problems is via profile likelihood methods (*Raue et al., 2009*; *Raue et al., 2010*).

Systems biology models may be computationally expensive to evaluate, making standard MCMC algorithms impractical as they require many evaluations. For our class of dynamic computational models the likelihood of each sample is based on the time integration of the model, which can be very demanding for large or stiff models. To confine sampling and integration costs, model order reduction techniques may be of help to speed up analysis (see Supplemental Information). However, the reduction errors affect prediction uncertainty differs per model and per prediction.

An alternative approach is to locally approximate the likelihood function using parameter sensitivities. Applications in systems biology of the latter idea include a gene network model in a Drosophila embryo (*Ashyraliyev, Jaeger & Blom, 2008*), a microbial growth model (*Schittkowski, 2007*), and an in-silico gene regulatory network (*Zak et al., 2003a*). This method is simple and has a very low computational demand since it avoids sampling, but may lead to a highly erroneous estimation of prediction uncertainty (*Gutenkunst et al., 2007a*), as we demonstrated (Fig. 2D). A better alternative is to use emulators and use adaptive error modelling within the MCMC algorithm (*Cui, Fox & O'Sullivan, 2011*).

In this paper the analysis is carried out with noiseless data. The Hessian matrix has full rank for each model, so all models are theoretically locally identifiable in the neighbourhood of the penalized maximum likelihood parameter. However, in practice the PML parameter might be hard to find or be non-unique due to noisy data. An alternative is to define $Q$ with respect to the average time course of the log-predictions (that is, averaging across the sample of parameter values), as is mentioned in the end of the Methods section. With noisy data, the uncertainty will increase compared to noiseless data, but the range of uncertainties across different predictions, as expressed by quantifiers such as $Q$, is likely to be similar to that in Fig. 3.

For biologists it is not always essential to obtain quantitative predictions and a reviewer asked about the uncertainty analysis of qualitative predictions. For this, the first step is to convert any quantitative predicted time course into a qualitative prediction, such as the statement that a certain metabolite or flux is going up, when a gene is over-expressed. Once this step has been decided upon, the prediction for any particular set of parameter values in the posterior sample is then either a simple *yes* (1) or *no* (0) or a value between 0 and 1 that measures the degree to which a single predicted time course corresponds with the statement. The second step is then to determine the mean and variance of the values obtained across the parameter sets in the posterior sample, which measure the extent to which the model predicts what is stated, and the prediction uncertainty, respectively. Note that for crisp (0/1) outcomes the mean is simply the fraction $f$ of the parameter sets that yields a *yes*, and the variance is $f(1-f)$.

## CONCLUSIONS

Our survey of a variety of models shows that prediction uncertainty is hard to predict. It turns out that it is practically impossible to establish the predictive power of a model without a full computational uncertainty analysis of each individual prediction. In practice this has strong consequences for the way models are transferred via the literature and databases. We conclude that publication of a model in the open literature or a database should not only involve the listing of the model equations, the parameter values with or without confidence intervals or parameter sensitivities, but should also include a sample of, say, 1000 draws representing the full (posterior) distribution of the parameters. Only then prospective users are able to reliably perform a full computational uncertainty analysis if they intend to use the model for prediction purposes. To this end, a software package is made available online, including an exemplary parameter sample. In addition we encourage to include experimental conditions, model assumptions, and used prior knowledge in order to make the sample reproducible. This will decrease the chances of error propagation, for example due to programming errors.

## ACKNOWLEDGEMENTS

The authors would like to thank the reviewers for their comments and W Kruijer, H van der Voet, S Schnabel, and E Boer for useful discussions.

### Funding

This work was supported by the Netherlands Consortium for Systems Biology (NCSB) which is part of the Netherlands Genomics Initiative/Netherlands Organisation for Scientific Research. The funders had no role in study design, data collection and analysis, decision to publish, or preparation of the manuscript.

### Grant Disclosures

The following grant information was disclosed by the authors:

The Netherlands Consortium for Systems Biology (NCSB).

### Competing Interests

Cajo ter Braak is an Academic Editor for PeerJ. Simon van Mourik and Jaap Molenaar are co-financed by the Netherlands Consortium for Systems Biology.

### Author Contributions

- Simon van Mourik contributed reagents/materials/analysis tools, wrote the paper, prepared figures and/or tables, reviewed drafts of the paper, software and literature study.
- Cajo ter Braak contributed reagents/materials/analysis tools, wrote the paper, reviewed drafts of the paper, software and literature study.
- Hans Stigter and Jaap Molenaar contributed reagents/materials/analysis tools, wrote the paper, reviewed drafts of the paper, literature study.

### Supplemental Information

Supplemental information for this article can be found online at http://dx.doi.org/10.7717/peerj.433.

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
