# Peer review of "Prediction uncertainty assessment of a systems biology model requires a sample of the full probability distribution of its parameters"

_PeerJ, doi:10.7717/peerj.433_

## Round 0.1 · original submission · Major Revisions

Both reviewers had specific comments. In particular the second reviewer raised several important points that you should address in your revision.

·

Basic reporting

No comments

Experimental design

No comments

Validity of the findings

No comments

Additional comments

The manuscript of Van Mourik et al. describes a methodology that allows to study the predictive quality of biological multi-parameter models by a full computational uncertainty analysis. They show that the predictive quality of sloppy biological models does not necessarily depends on the parameter uncertainty. Furthermore they show that the quality of predictions vary within one model and that this depends on the type of prediction. The methodology is very relevant since it gives systems biologist the opportunity to test their biological models for predictive quality for different predictions. However, I have some questions.

• For biologists it is not always essential to obtain quantitative predictions. The method described in the manuscript results in a quantifier, which is a degree of how accurate the predictions are quantitatively as far as I understood. However, in some cases a qualitative prediction can be enough to obtain new insights in how a biological system works. For instance it might be enough to know if a certain metabolite or flux is going up without an accurate prediction on how much the metabolite/flux goes up. Would it be possible with this method to obtain predictive values for qualitative predictions instead of quantitative predictions?

• The method shows that including data points can increase the prediction quality of the biological model. An experimentalist would like to know what kind of data/experiments are needed to increase the predictive quality. Would it be possible to use the method to determine what kind of data is needed to increase the predictive quality?

• In the last paragraph of the conclusion the authors state that a software package is made available. It would be useful to notify where we can find this software package (I could not find it, but maybe I overlooked it).

Reviewer 2 ·

Basic reporting

As far as I'm concerned the article addresses all areas that are associated to Basic Reporting's criteria. I would however like to suggest a few changes and point out some of the mistakes in the text that will boost up the readability of the text.

1) I do not find the choice of two text books that are cited with respect
to the Bayesian framework very suitable for this subject. A book from
Nate Silver (2012) is a popular science book, which is, apart from being
a good read, has no pedagogical value for learning statistics
and Bayesian inference. Another book, McGrayne (2011), is a historical
assessment on the importance of Bayes' theorem, which again is not meant
to be a text book for learning the subject. I suggest instead authors
refer to other valuable, and scienti cally sound, texts for this subject.
There are several excellent texts, such as a book from Edwin Jaynes titled
Probability Theory As Extended Logic, or David MacKay's "Information
theory, inference and learning algorithms", or "Data analysis: a Bayesian
tutorial\ by Sivia and Skilling.

2) At line 317, for the acronym GLUE, it is not de ned that what is it exactly
standing for.

3) In line 21 there is one extra "for".

4) The title suggests that characterizing model's prediction quality needs
a "full" probability distribution. This is a valid suggestion, but this work does
not taking into account the entire probability distribution of parameters.
Whereas, a sample of parameters (1000 samples) drawn by a MCMC algorithm
is considered. It is of course not a single parameter fit, like maximum
likelihood estimation, but it is not also the entire distribution of parameter
space.

5) At line 57, for the frst time authors mention MCMC algorithms and
provide a definition to it. I fi nd the definition, although correct, but rather
narrow. MCMC algorithms have broader applications such as estimating
posteriors, or numerically calculating an integral which is cumbersome to
calculate analytically. Hence I would suggest to put the MCMC approach
into a broader context and subsequently narrow it down to its particular
application for the current problem.

6) Just as a suggestion, although several texts refer to the uniform prior distribution
as "uninformative", I find it an incorrect usage of the term. The uniform probability still contains information, such as the boundaries of the parameter, or even saying that "every value of parameter within the boundary is equally likely" contains information, in my opinion. Therefore, maybe using other terms such as ignorant prior or even uniform prior is a better practice.

7) In line 303, authors mentioned that the prior distribution in Bayesian
analysis is a means to ensure that a parameter does not lose its physical
meaning at parameter fitting; I am not sure if I understand this statement,
so please clarify it further.

Experimental design

Authors suggested a quantity Q(theta) that is basically the log-ratio of the
model's prediction under parameter to the real value (or the model's prediction
under maximum likelihood parameter estimation (MLE), across time. A distribution
of Q(theta) is calculated, for a preferably large number of samples of parameters 
drawn by a MCMC algorithm. With regard to this technique for characterizing
prediction quality I have several questions that would like to be addressed by
the authors.

1) How the equation for Q(theta) is derived? Under what assumptions such a
formula can be obtained?

2) I suppose this concept relies on a choice of "correct" prediction, from
which the dispersion of model's predictions is quantfi ed when the model's
parameters are perturbed. Here it is suggested that the predictions of the
model under maximum likelihood parameter MLE represent the correct
prediction. How much this choice is justified under different models? Under
what conditions, and assumptions, the model that is trained by
MLE provides the correct prediction?

3) Looking at the figure 2C and 2D, they resemble the figure 4B from the
reference (Gutenkunst 2007b). This raises a question that how novel is the
proposed quantity Q. If they are based on diff erent assumptions, it can
be useful to clarify this explicitly in the paper. Plus, I think a comparison
between these two approaches is necessary to pin down diff erences clearly.

4) It is described that the LCA method is also considered for characterizing
prediction uncertainty, I fi nd it useful to mention and consider MCA
method as well, which can be found in (Gutenkunst 2007a). This again
boils down to comparison of the proposed quantity Q and Gutenkunst et
al.

5) In the text, I found it several times that authors used the term "dimensionless"
for Q (e.g., lines 237 and 128), as it is an extra advantage for Q, when
compared to other approaches. If it is important to highlight the so-called
dimensionless quality of Q, it can be useful to discuss why it is important
and whether other methods are lacking this quality.

6) In section "Estimation of the posterior distribution of the parameters",
the general parameter estimation under Bayesian framework is discussed.
Bayesian framework enjoys incorporating prior knowledge on the acceptable parameter's value, and the current observed data, into a unified
framework for parameter estimation. In the text, authors used a uniform
distribution for the prior distribution which is for situations that given
our information we are ignorant about the parameter's value. Firstly, it
might be constructive if authors explicitly discuss the different choices of
the prior distribution.

7) If an ignorant (uniform) prior is replaced by another prior such as Gaussian
prior or other members of exponential family, how this will be translated
into the sloppy parameter sensitivity. And consequently, how much this
will affect the prediction quality Q of the model. Does this boost the
model's prediction quality or does not change the outcome?

Validity of the findings

1) It is mentioned in discussion (line 346) that the proposed approach is
applied on noiseless data, and where finding MLE is computationally
easy. However, once fi nding MLE is difficult, authors suggested to take
the average predictions. First, average predictions under what choice of theta. Second, how this point is formally justified?

2) I suggest to have an extra section in the main text to discuss the application
of this approach on noisy data. And discuss different scenarios that
might be faced when dealing with a data that is contaminated with various
noise components. Therefore, the authors can justify different choices
for when MLE is difficult to determine.

3) At line 364, authors concluded that the publication of models should include
a number of parameter samples from the parameter space posterior,
which is an acceptable point. However, this has been suggested several
times before in other publications, such as reference (Brown 2003). If there is
any novelty in the suggestion, authors might consider elaborate this point
further, otherwise mention previous works that have pointed this out.

Additional comments

The proposed quantity Q in fact quantities the dispersion of predictions
around a reference point, that is proposed to be fixed at the predictions from the
maximum likelihood estimation (MLE) parameters. Therefore, the quality of
predictions is implicitly defined as how much predictions diverge from a reference
point. In Bayesian framework, on the other hand, a credible interval is defined as
an interval that comprises a given amount of posterior distribution. The credible
(and con fidence) intervals are associated to the "certainty" of predictions, and
the stability of predictions in proximity of a reference point is not of concern. In
the text, terms "certainty" and "quality" (as it is characterized by Q) are used
interchangeably. However, quality of prediction is not a well-de fined concept,
but the certainty of predictions are rigorously de fined in relevant literature. All
in all, I did not fi nd the argument that Q quantfi es the certainty in prediction
convincing, it is merely a measure of dispersion around a single reference point.
Therefore, I would like to ask authors to clarify this point, otherwise it would
lead to confusion and misconception for the readers.

Based on the definition of sloppy parameter sensitivity, it is logically easy
to see that under variation of parameters along the sloppy axis the model's
prediction does not change dramatically. Therefore, some would mistake
this as the robustness in model. Model's with a high-degree of sloppiness
would exhibit a more robust character. However, there is another concept
where the certainty in predictions that when a parameter estimation is
less certain, the prediction is less trustworthy. The proposed quantity
Q actually characterizes the former, the variation of model's prediction,
or loosely de fined robustness. I believe it is more useful to quantify the
model's prediction certainty. Or it must be explicitly discussed in the text.

---

## Round 0.2 · accepted · Accept

Thank you for carefully revising your manuscript.

·

Basic reporting

No Comments

Experimental design

No Comments

Validity of the findings

No Comments

Additional comments

The revised manuscript is improved substantially and the authors addressed all the questions satisfactorily.

One minor point:
- line 134: there is twice 'biologists are' in succession